# Learning Macroscopic Brain Connectomes via Group-Sparse Factorization

Farzane Aminmansour[1], Andrew Patterson[1], Lei Le[2], Yisu Peng[3], Daniel Mitchell[1], Franco Pestilli[4],
Cesar Caiafa[5,6], Russell Greiner[1] and Martha White[1]

[1]Department of Computing Science, University of Alberta, Edmonton, Alberta, Canada
[2]Department of Computer Science, Indiana University, Bloomington, Indiana, USA
[3]Department of Computer Science, Northeastern University, Boston, Massachusetts, USA
[4]Department of Psychological and Brain Sciences, Indiana University, Bloomington, Indiana, USA
[5]Instituto Argentino de Radioastronomía- CCT La Plata, CONICET / CIC-PBA, V. Elisa, Argentina
[6]Tensor Learning Unit- Center for Advanced Intelligence Project, RIKEN, Tokyo, Japan
{aminmans, ap3, daniel7, rgreiner whitem}@ualberta.ca, {leile}@iu.edu,
{peng.yis}@husky.neu.edu, {franpest}@indiana.edu, {ccaiafa}@gmail.com

## Abstract

Mapping structural brain connectomes for living human brains typically requires expert analysis and rule-based models on diffusion-weighted magnetic resonance imaging. A data-driven approach, however, could overcome limitations in such rule-based approaches and improve precision mappings for individuals. In this work, we explore a framework that facilitates applying learning algorithms to automatically extract brain connectomes. Using a tensor encoding, we design an objective with a group-regularizer that prefers biologically plausible fascicle structure. We show that the objective is convex and has a unique solution, ensuring identifiable connectomes for an individual. We develop an efficient optimization strategy for this extremely high-dimensional sparse problem, by reducing the number of parameters using a greedy algorithm designed specifically for the problem. We show that this greedy algorithm significantly improves on a standard greedy algorithm, called Orthogonal Matching Pursuit. We conclude with an analysis of the solutions found by our method, showing we can accurately reconstruct the diffusion information while maintaining contiguous fascicles with smooth direction changes.

## 1 Introduction

A fundamental challenge in neuroscience is to estimate the structure of white matter connectivity in the human brain or connectomes [14, 29]. Connectomes are made up of neuronal axon bundles wrapped with myelin sheaths, called fascicles, and connect different areas of the brain. Acquiring information about brain tissue is possible by measuring the diffusion of water molecules at different spatial directions. Fascicles can be inferred by employing tractography algorithms, which calculate mathematical models from the diffusion-weighted signal. Currently, diffusion-weighted magnetic resonance imaging (dMRI) combined with fiber tractography is the only method available to map structural brain connectomes in living human brains [3, 30, 23]. This method has revolutionized our understanding of the network structure of the human brain and the role of white matter in health and disease.

Standard practice in mapping connectomes is comprised of several steps:a dMRI is acquired (Fig 1A), a model is fit to the signal in each brain voxel (Fig. 1B) and a tractography algorithm is used to estimate long range brain connections (Fig. 1C). Multiple models can be used at each one of these

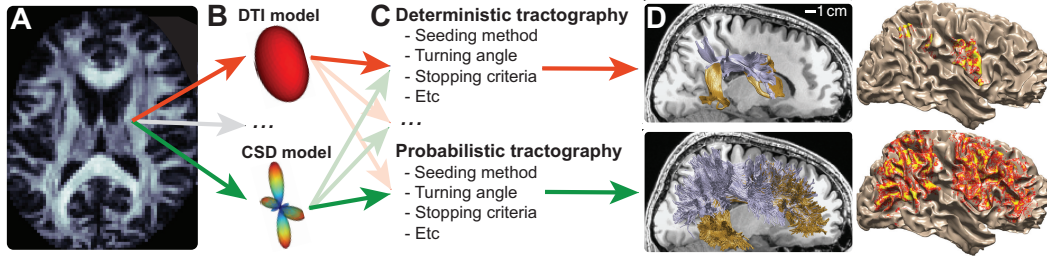

Figure 1: **A**: Measurements of white matter using diffusion-weighted magnetic resonance imaging (dMRI). **B**: Multiple models can describe the dMRI signal in each brain voxel. For example, the diffusion-tensor model (DTI; top, [2]) and the constrained-spherical deconvolution model (CSD, bottom; [28]) are commonly used. **C**: Multiple tractography methods integrate model fits across voxels to estimate long-range brain connections. There are many tractography algorithms exist, each with multiple parameters, for both deterministic and probabilistic methods [27]. In principle several combinations of methods and parameters are used by investigators. **D**: **Left**: Two major white matter tracts, the Arcuate Fasciculus in gold and superior lateral fasciculus in lilac, reconstructed in a single brain using deterministic (top) and probabilistic (bottom) tractography. **Right**: Cortical termination of the superior lateral fasciculus in the same brain estimated with deterministic (top) and probabilistic (bottom) tractography. Arrows show multiple possible choices of model and parameters to generate connectome estimates (D) from dMRI data (A).

steps and each model allows multiple parameters to be set. Currently, best practice in the field is to choose one model and pick a single set of parameters using heuristics such as recommendations by experts or previous publications. This rule-based approach has several limitations. For example, different combinations of models and parameters generate different solutions (Fig 1D). Figure 1 exemplifies how from a single dMRI data set collected in a brain, choosing a single model and parameters set (Fig. 1A-C) can generate vastly different connectome mapping results (Fig 1D; adapted from [20]). In the figure, we show that both estimates of white matter tracts (Fig 1D left) and cortical connections (Fig. 1D right) vary substantially even within a single brain.

There have been some supervised learning approaches proposed for tractography. These supervised methods, however, such as those using random forests [17] and neural networks [22, 5] require labelled data. This means tractography solutions must first be given for training, limiting the models mainly to mimic expert solutions rather than learn structures beyond them. A few methods have used regularized learning strategies, but for different purposes, such as removing false connections in the given tractography solution [12] and using radial regularization for micro-structure [9].

This work presents a fully unsupervised learning framework for tractography. We exploit a recently introduced encoding for connectome data, called ENCODE [8], which represents dMRI (and white matter fascicles) as a tensor factorization. This factorization was previously used only to represent expert connectomes as a tensor, generated using a standard rule-based tractography process introduced in Fig. 1. We propose to instead learn this tensor using the dMRI data, to learn the structure of brain connectomes. We introduce a regularized objective that attempts to extract a tensor that reflects a biologically plausible fascicle structure while also reconstructing the diffusion information. We address two key challenges: (1) designing regularizers that adequately capture biologically plausible tract structures and (2) optimizing the resulting objective for an extremely high-dimensional and sparse tensor. We develop a group regularizer that captures both spatial and directional continuity of the white matter fascicles. We solve this extremely high-dimensional sparse problem using a greedy algorithm to screen the set of possible solutions upfront. We prove both that the objective is convex, with a unique solution, and provide approximation guarantees on the greedy algorithm. We then show that this greedy algorithm much more effectively selects possible solutions, as compared to a standard greedy algorithm called Orthogonal Matching Pursuit (OMP). We show, both quantitatively and qualitatively, that the solutions provided by our method effectively reconstruct the diffusion information in each voxel while maintaining contiguous, smooth fascicles.

---

The code is available at: https://github.com/framinmansour/Learning-Macroscopic-Brain-Connectomes-via-Group-Sparse-Factorization

## 2   Encoding Brain Connectomes as Tensors

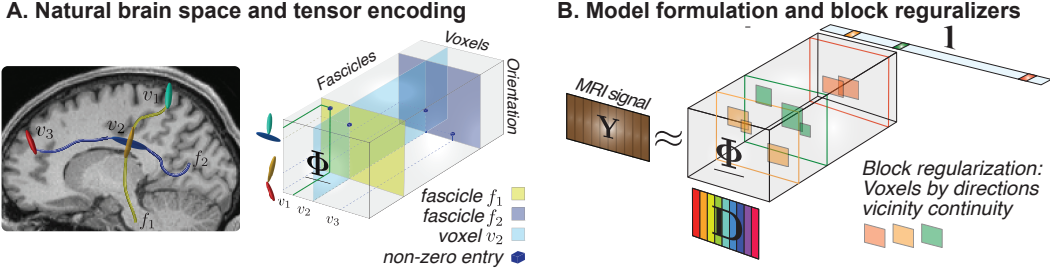

**A. Natural brain space and tensor encoding**

**B. Model formulation and block regularizers**

Figure 2: **A**: The ENCODE method; from natural brain space to tensor encoding. **Left**: Two example white matter fascicles ($f_1$ and $f_2$) passing through three voxels ($v_1$, $v_2$ and $v_3$). **Right**: Encoding of the two fascicles in a three dimensional tensor. The non-zero entries in $\underline{\mathbf{\Phi}}$ indicate fascicles orientation (1st mode), position (voxel, 2nd mode) and identity (3rd mode). **B**: Model formulation and group sparse regularization. Depiction of how ENCODE facilitates integration of dMRI signal, $\mathbf{Y}$, connectome structure, $\underline{\mathbf{\Phi}}$, and a dictionary of predictions of the dMRI signal, $\mathbf{D}$, for each fascicle orientation. The group regularizers (orange and green squares) defines pairwise groups of neighbouring voxels and similar orientations. Note that the voxels are linearized to enable $\underline{\mathbf{\Phi}}$ and the groups to be visualized. This allows us to flatten four-dimensional hyper-cubes—three dimensions for voxels and one for orientations—to squares.

ENCODE [8] maps fascicles from their natural brain space into the three dimensions of a sparse tensor $\underline{\mathbf{\Phi}} \in \mathbb{R}^{N_a \times N_v \times N_f}$ (Fig. 2A - right). The first dimension of $\underline{\mathbf{\Phi}}$ (1st mode, size $N_a$) encodes individual white matter fascicles orientation at each position along their path through the brain. Individual segments (nodes) in a fascicle are coded as non-zero entries in the sparse array (dark-blue cubes in Fig. 2A - right). The second dimension of $\underline{\mathbf{\Phi}}$ (2nd mode, size $N_v$) encodes fascicles spatial position within the voxels of dMRI data. Slices in this second dimension represent single voxels (cyan slice in Fig. 2A - right). The third dimension (3rd mode, size $N_f$) encodes the indices of each fascicle within the connectome. Full fascicles are encoded as $\underline{\mathbf{\Phi}}$ frontal slices (cf., yellow and blue in Fig. 2A - right). Within one tract, such as the Arcuate Fasciculus, the model we use has fine-grained orientations $N_a = 1057$, with number of fascicles $N_f = 868$ and number of voxels $N_v = 11,823$.

ENCODE facilitates the integration of measured dMRI signals with the connectome structure (Fig. 2B - right). DMRI measurements are collected with and without a diffusion sensitization magnetic gradient and along several gradient directions or $N_\theta$, i.e. $\theta \in R^3$. In the Arcuate Fasciculus for instance, the data was collected for $N_\theta = 96$ different angles of gradient direction. Then, the dMRI signal is represented as matrix $\mathbf{Y} \in \mathbb{R}^{N_\theta \times N_v}$, which represents the value of diffusion signal received from each voxel when any individual angle of gradient directions were applied during the scanning.

Moreover, ENCODE allows factorizing the dMRI signal as the product of a 3-dimensional tensor $\underline{\mathbf{\Phi}} \in \mathbb{R}^{N_a \times N_v \times N_f}$ and a dictionary of dMRI signals $\mathbf{D} \in \mathbb{R}^{N_\theta \times N_a}$: $\mathbf{Y} \approx \underline{\mathbf{\Phi}} \times_1 \mathbf{D} \times_3 \mathbf{1}$. The notation "$\times_n$" is the tensor-by-matrix product in mode-$n$ (see [15]). The dot product with $\mathbf{1} \in \mathbb{R}^{N_f}$ sums over the fascicle dimension.[1] The matrix $\mathbf{D}$ is a dictionary of representative diffusion signals: each column represents the diffusion signal we expect to receive from any axon in the direction of any possible fascicle orientation $a$ by sensitizing magnetic gradient in each direction of $\theta$. More specifically, the entries are computed as follows: $\mathbf{D}(\theta, a) = e^{-b\theta^T Q_a \theta} - \frac{1}{N_\theta} \sum_\theta e^{-b\theta^T Q_a \theta}$, in which $Q_a$ is an approximation of diffusion tensor per fascicle-voxel and scalar $b$ denotes the diffusion sensitization gradient strength. $\theta^T Q_a \theta$ gives us the diffusion at direction $\theta$ generated by fascicle $f$.

## 3   A Tractography Objective for Learning Brain Connectomes

The original work on ENCODE assumed the tensor $\underline{\mathbf{\Phi}}$ was obtained from a tractography algorithm. In this section, we instead use this encoding to design an objective to learn $\underline{\mathbf{\Phi}}$ directly from dMRI

data. First consider the problem of estimating tensor $\underline{\mathbf{\Phi}}$ to best predict $\mathbf{Y}$, for a given $\mathbf{D} \in \mathbb{R}^{N_\theta \times N_a}$. We can use a standard maximum likelihood approach (see Appendix A for the derivation), to get the following reconstruction objective

$$\hat{\underline{\mathbf{\Phi}}} = \operatorname*{argmin}_{\underline{\mathbf{\Phi}} \in \mathbb{R}^{N_a \times N_v \times N_f}} \|\mathbf{Y} - \underline{\mathbf{\Phi}} \times_1 \mathbf{D} \times_3 \mathbf{1}\|_F^2, \tag{1}$$

where $\|\cdot\|_F$ is the Frobenius norm that sums up the squared entries of the given matrix. This objective prefers $\underline{\mathbf{\Phi}}$ that can accurately recreate the diffusion information in $\mathbf{Y}$. This optimization, however, is highly under-constrained, with many possible (dense) solutions.

In particular, this objective alone does not enforce a biologically plausible fascicle structure in $\underline{\mathbf{\Phi}}$. The tensor $\underline{\mathbf{\Phi}}$ should be highly sparse, because each voxel is expected to have only a small number of fascicles and orientations [20]. For example, for the Arcuate Fasciculus, we expect at most an activation level in $\underline{\mathbf{\Phi}}$ of $(N_v \times 10 \times 10/(N_a \times N_v \times N_f) \approx 1e{-}6$, using a conservative upper bound of 10 fascicles and 10 orientations on average per voxel. Additionally, the fascicles should be contiguous and should not sharply change orientation.

We design a group regularizer to encode these properties. Anatomical consistency of fascicles is enforced locally within groups of neighboring voxels and orientations. Overlapping groups are used to encourage this local consistency to result in global consistency. Group regularization prefers to zero all coefficients for a group. This zeroing has the effect of clustering non-zero coefficients in local regions within the tensor, ensuring similar fascicles and orientations are active based on spatial proximity. Further, overlapping groups encourages neighbouring groups to either both be active or inactive for a fascicle and direction. This promotes contiguous fascicles and smooth direction changes. These groups are depicted in Figure 2B, with groups defined separately for each fascicle (slice). We describe the group regularizer more formally in the remainder of this section.

Assume we have groups of voxels $\mathcal{G}_\mathcal{V} \in \mathcal{V}$ based on spatial coordinates and groups of orientations $\mathcal{G}_\mathcal{A} \in \mathcal{A}$ based on orientation similarity. For example, each $\mathcal{G}_\mathcal{V}$ could be a set of 27 voxels in a local cube; these cubes of voxels can overlap between groups, such as $\{(1,1,1),(1,1,2),\ldots,(3,3,3)\} \in \mathcal{V}$ and $\{(2,1,1),(2,1,2),\ldots,(4,3,3)\} \in \mathcal{V}$. Each $\mathcal{G}_\mathcal{A}$ can be defined by selecting one atom (one orientation) and including all orientations in the group that have a small angle to that central atom, i.e., an angle that is below a chosen threshold. Consider one orientation, voxel, fascicle triple $(a, v, f)$. Assume a voxel has a non-zero coefficient for a fascicle: $\underline{\mathbf{\Phi}}_{a,v,f}$ is not zero for some $a$. A voxel within the same group $\mathcal{G}_\mathcal{V}$ is likely to have the same fascicle with a similar orientation. A distant voxel, on the other hand, is highly unlikely to share the same fascicle. The goal is to encourage as many pairwise groups $(\mathcal{G}_\mathcal{V}, \mathcal{G}_\mathcal{A})$ to be inactive—have all zero coefficients for a fascicle—and concentrate activation in $\underline{\mathbf{\Phi}}$ within groups.

We can enforce this group sparsity by adding a regularizer to (1). Let $x_{\mathcal{G}_\mathcal{A}, v, f} \in \mathbb{R}$ indicate whether a fascicle $f$ is active for voxel $v$, for any orientation $a \in \mathcal{G}_\mathcal{A}$. Let $\mathbf{x}_{\mathcal{G}_\mathcal{A}, \mathcal{G}_\mathcal{V}, f}$ be the vector composed of these identifiers for each $v \in \mathcal{G}_\mathcal{V}$. Either we want the entire vector $\mathbf{x}_{\mathcal{G}_\mathcal{A}, \mathcal{G}_\mathcal{V}, f}$ to be zero, meaning the fascicle is not active in any of the voxels $v \in \mathcal{G}_\mathcal{V}$ for the orientations $a \in \mathcal{G}_\mathcal{A}$. Or, we want more than one non-zero entry in this vector, meaning multiple nearby voxels share the same fascicle. This second criterion is largely enforced by encouraging as many blocks to be zero as possible, because each voxel will prefer to activate fascicles and orientations in already active pairs $(\mathcal{G}_\mathcal{V}, \mathcal{G}_\mathcal{A})$. As with many sparse approaches, we use an $\ell_1$ regularizer to set entire blocks to zero. In particular, as has been previously done for block sparsity [26], we can use an $\ell_1$ across the blocks $\mathbf{x}_{\mathcal{G}_\mathcal{A}, \mathcal{G}_\mathcal{V}, f}$

$$\sum_{f \in \mathcal{F}} \sum_{\mathcal{G}_\mathcal{V} \in \mathcal{V}} \sum_{\mathcal{G}_\mathcal{A} \in \mathcal{A}} \|\mathbf{x}_{\mathcal{G}_\mathcal{A}, \mathcal{G}_\mathcal{V}, f}\|_2. \tag{2}$$

The outer sums can be seen as an $\ell_1$ norm across the vector of norm values containing $\|\mathbf{x}_{\mathcal{G}_\mathcal{A}, \mathcal{G}_\mathcal{V}, f}\|_2$. This encourages $\|\mathbf{x}_{\mathcal{G}_\mathcal{A}, \mathcal{G}_\mathcal{V}, f}\|_2 = 0$, which is only possible if $\mathbf{x}_{\mathcal{G}_\mathcal{A}, \mathcal{G}_\mathcal{V}, f} = \mathbf{0}$.

Finally, we need to define a continuous indicator variable $\mathbf{x}_{\mathcal{G}_\mathcal{A}, \mathcal{G}_\mathcal{V}, f}$ to simplify the optimization. A 0-1 indicator is discontinuous, and would be difficult to optimize. Instead, we use the following continuous indicator

$$\mathbf{x}_{\mathcal{G}_\mathcal{A}, \mathcal{G}_\mathcal{V}, f} = [\|\underline{\mathbf{\Phi}}_{\mathcal{G}_\mathcal{A}, v_1, f}\|_1, \ldots, \|\underline{\mathbf{\Phi}}_{\mathcal{G}_\mathcal{A}, v_n, f}\|_1] \quad \text{for each } v_i \in \mathcal{G}_\mathcal{V} \tag{3}$$

An entry in $\mathbf{x}_{\mathcal{G}_\mathcal{A}, \mathcal{G}_\mathcal{V}, f}$ is 0 if fascicle $f$ is not active for $(\mathcal{G}_\mathcal{V}, \mathcal{G}_\mathcal{A})$. Otherwise, the entry is proportional to the sum of the absolute coefficient values for that fascicle for orientations in $\mathcal{G}_\mathcal{A}$.

Our proposed group regularizer is

$$R(\underline{\mathbf{\Phi}}) = \sum_{f \in \mathcal{F}} \sum_{\mathcal{G}_\mathcal{V} \in \mathcal{V}} \sum_{\mathcal{G}_\mathcal{A} \in \mathcal{A}} \|\mathbf{x}_{\mathcal{G}_\mathcal{A}, \mathcal{G}_\mathcal{V}, f}\|_2 = \sum_{f \in \mathcal{F}} \sum_{\mathcal{G}_\mathcal{V} \in \mathcal{V}} \sum_{\mathcal{G}_\mathcal{A} \in \mathcal{A}} \sqrt{\sum_{v \in \mathcal{G}_\mathcal{V}} \left( \sum_{a \in \mathcal{G}_\mathcal{A}} |\underline{\mathbf{\Phi}}_{a,v,f}| \right)^2}, \quad (4)$$

which, combined with equation (1), gives our proposed objective. Given the observed $\mathbf{Y}$, and the dictionary $\mathbf{D}$, find the $\underline{\mathbf{\Phi}}$ s.t.

$$\min_{\underline{\mathbf{\Phi}} \in \mathbb{R}^{N_a \times N_v \times N_f}} \|\mathbf{Y} - \underline{\mathbf{\Phi}} \times_1 \mathbf{D} \times_3 \mathbf{1}\|_F^2 + \lambda R(\underline{\mathbf{\Phi}}) \quad (5)$$

for regularization weight $\lambda > 0$. This objective balances between reconstructing diffusion data and constraints on the structure in $\underline{\mathbf{\Phi}}$. Crucially, this objective is convex in $\underline{\mathbf{\Phi}}$ and has a unique solution, which we show in Theorem 1 in Appendix B. Uniqueness ensures identifiable tractography solutions and convexity facilitates obtaining optimal solutions.

## 4    An Efficient Algorithm for the Tractography Objective

Standard gradient descent algorithms can be used directly on (5) to find the optimal solution. Unfortunately, the number of parameters in the optimization is very large: $N_v \times N_f \times N_a$ is billions even for just one tract. At the same time, the number of active coefficients at the end of the optimization is much smaller, only on the order of $N_v$, because there are only handful of fascicles and orientations per voxel. Even when initializing $\underline{\mathbf{\Phi}}$ to zero, the gradient descent optimization might make all of $\underline{\mathbf{\Phi}}$ active during the optimization. Screening algorithms have been developed to prune entries for sparse problems [31, 6]. These generic methods, however, still have too many active coefficients to make this optimization tractable for wide application, as we have verified empirically.

Instead, we can design a screening algorithm specialized to our objective. Orientations can largely be selected independently for each voxel, based solely on diffusion information. We can infer the likely orientations of fascicles in a voxel that could plausibly explain the diffusion information, without knowing precisely which fascicles are in that voxel. If we can select a plausible set of orientations for each voxel before optimizing the objective, we can significantly reduce the number of parameters. For example, 20 orientations is a large superset, but would reduce the number of parameters by a factor of 10,000 because the whole $N_a = 120,000$.

One strategy is to generate these orientations greedily, such as with a method like Orthogonal Matching Pursuit (OMP). This differs from most screening approaches, which usually iteratively prune starting from the full set. Generating orientations starting from an empty set, rather than pruning, is a more natural strategy for such an extremely sparse solution, where only 0.017% of the items are used. Consider how OMP might generate orientations. For a given voxel $v$, the next best orientation is greedily selected based on how much it reduces the residual error for the diffusion. On the first step, it adds the single best orientation for predicting the $N_\theta = 96$ dimensional diffusion vector for voxel $v$. It generates up to a maximum of $k$ orientations greedily and then stops. Then only coefficients for this set of orientations will be considered for voxel $v$ in the optimization of the tractography objective. This procedure is executed for each voxel, and is very fast.

Though a greedy strategy for generating orientations is promising, the criterion used by OMP is not suitable for this problem. Using residual errors for the criterion prefers orthogonal or dissimilar orientations, to provide a basis with which to easily reconstruct the signal. The orientations in voxels, however, are unlikely to be orthogonal. Instead, it is more likely that there are multiple fascicles with similar orientations in a voxel, with some fascicles overlapping in a different—but not necessarily orthogonal—direction. We must modify the selection criterion to select a number of similar orientations to reconstruct the diffusion information in a voxel.

To do so, we rely on the more general algorithmic framework for subselecting items from a set, of which OMP is a special case. We need to define a criterion which evaluates the quality of subsets $S$ from the full set of items $\mathcal{S}$. In our setting, $\mathcal{S}$ is the full set of orientations and $S$ a subset of those orientations. Our goal is to find $S \subset \mathcal{S}$ with $|S| \leq k$ such that $\bar{g}(S)$ is maximal. If we can guarantee this criterion $\bar{g} : \mathcal{P}(\mathcal{S}) \to \mathbb{R}$ is (approximately) submodular, then we can rely on a wealth of literature showing the effectiveness of greedy algorithms for picking $S$ to maximize $\bar{g}$.

We use a simple modification on the criterion for OMP, the $g(S) =$ the squared multiple correlation [13]. We propose a simple yet effective modification, and define the Orientation Greedy criterion as

$$\bar{g}(S) \stackrel{\text{def}}{=} g(S) + \sum_{s \in S} g(\{s\})$$

This objective balances between preferring a set $S$ with high multiple correlation, and ensuring that each orientation itself is useful. Each orientation likely explains a large proportion of the diffusion for a voxel. This objective will likely prefer to pick two orientations that are similar that recreate the diffusion in the voxel well. This contrasts two orthogonal orientations, that can be linearly combined to produce those two orientation but that themselves do not well explain the diffusion information. This modification is conceptually simple, yet now has a very different meaning. The simplicity of the modification is also useful for the optimization, since a linear sum of submodular functions is itself again submodular. We provide approximation guarantees for this submodular maximization in Appendix D, using results for the multiple correlation [13].

The full algorithm consists of two key steps. The first step is to screen the orientations, using Orientation Greedy in Algorithm 1. We then use subgradient descent to optimize the Tractography Objective using this much reduced set of parameters. The second step prunes this superset of possible orientations further, often to only a couple of orientations. The resulting solution only has a small number of active fascicles and orientations for each voxel. We provide a detailed derivation and description of the algorithm in Appendix C.

The optimization given the screened orientations remains convex. The main approximation in the algorithm is introduced from the greedy selection of orientations. We provide approximation guarantees for how effectively the greedy algorithm maximizes the criterion $\bar{g}$. But, this does not characterize whether the criterion itself is a suitable strategy for screening. In the next section, we focus our empirical study on the efficacy of this greedy algorithm, which is critical for obtaining efficient solutions for the tractography objective.

## 5 Empirical results: Reconstructing the anatomical structure of tracts

We investigate the properties of the proposed objective on two major structures in the brain. The first is the Arcuate Fasciculus, hereafter Arcuate. The other is the Arcuate combined with one branch of the Superior Longitudinal Fasciculus, SLF1, hereafter ARC-SLF. Due to space constraints, we relegate additional empirical results on ARC-SLF to Appendix E.6. We learn on data generated by an expert connectome solution within the ENCODE model (Appendix E.2). This allows us to objectively investigate the efficacy of the objective and greedy optimization strategy, because we have access to the ground truth $\underline{\Phi}$ that generated the data. To the best of our knowledge, this is the first completely unsupervised data-driven approach for extracting brain connectomes. We, therefore, focus primarily on understanding the properties of our learning approach for tractography.

We particularly (a) investigate how effectively our Greedy algorithm selects orientations, (b) investigate how accurately the group regularized objective with this screening approach can reconstruct the diffusion information, and (c) visualize the plausibility of the solutions produced by our method, particularly in terms of smoothness of the fascicles. Even with screening, this optimization when learning over all fascicles and voxels, is prohibitively expensive for running thorough experiments. We therefore focus first on evaluating the model given the assignment of fascicles to voxels, meaning for the following experiments fascicles are fixed. Because the largest approximation in the algorithm is the greedy selection of orientations, this is the most important step to understand first. For a given set of (greedily chosen) orientations, the objective remains convex with a unique solution. We know, therefore, that further optimizing over fascicles as well would only reduce the reconstruction error.

### 5.1 Screening

We define two error metrics to demonstrate the utility of GreedyOrientation over OMP for this task. The first is the total number of orientations present in $\underline{\Phi}$-expert that are not present in $\underline{\Phi}$ generated by the screening approach, measuring the exactness of the solution. The second metric is the minimum possible angular distance between each of the orientations in $\underline{\Phi}$-expert with any arbitrary set of orientations in the corresponding voxel of $\underline{\Phi}$ generated by the screening approach, so that the set

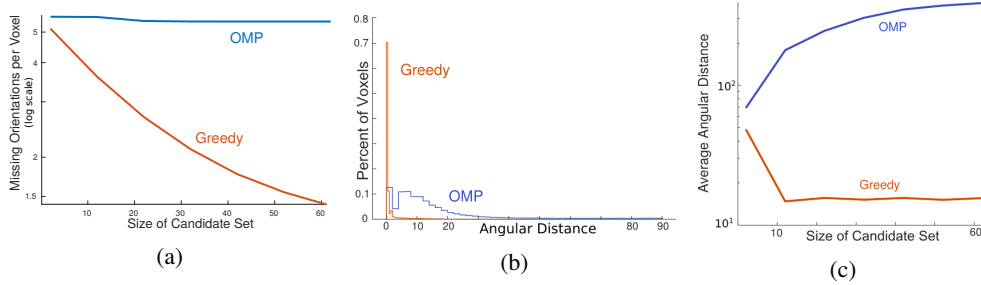

(a)                    (b)                    (c)

Figure 3: **(a):** Average number of missing orientations per voxel in candidate sets of increasing size. **(b):** The distribution of angular distances from the ground truth of OMP and GreedyOrientation after global optimization procedure. The angular distance is the minimum possible distance given some weighted combination of selected orientations. **(c):** Average angular distance between the weighted sum of predicted node orientations and the ground truth in each voxel for candidate sets of increasing size.

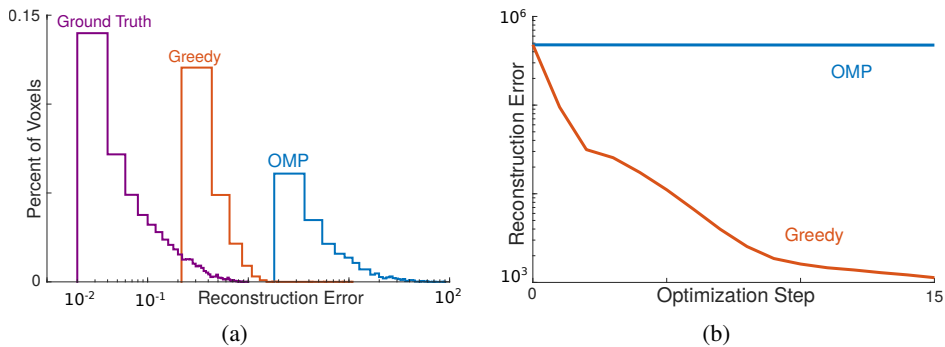

(a)                                  (b)

Figure 4: **(a):** Comparing the distribution of reconstruction error for ground truth, OMP, and GreedyOrientation over voxels after optimization. **(b):** The improvement of reconstruction error during the steps of gradient decent shows that the objective is not able to improve the OMP selected orientation sets while it is improving the GreedyOrientation choices constantly.

would provide the best possible approximation of that orientation. The details of algorithm can be found in Appendix E.5.

We demonstrate the screening method's performance using both error metrics in Figure 3. In Figure 3a, we show the effect of increasing the size of our candidate set of orientations on the number of missing orientations compared to the ground truth. GreedyOrientation's advantage is likely because OMP continually adds dissimilar orientations, thus is less likely to add the exactly correct orientations because these are too similar to orientations already in the candidate set. Figure 3b shows the minimum angular distance given a linear combination of orientations in the candidate set compared to the ground truth. GreedyOrientation has high probability mass near zero, showing that it generates appropriate candidate sets. Finally, Figure 3c shows that the angular distances between the orientations weighted with the optimized weights and ground truth for different size of orientations candidate set.

We can clearly see that increasing the size of the orientation set in OMP results in a larger angular distance since more dissimilar orientations are included. On the other hand, the angular distance of candidate sets chosen by GreedyOrientation decreases fast and then stabilized, which indicates that GreedyOrientation forward selection criterion is defined well so that the best candidate orientations approximate the ground truth are among the immediate ones. Moreover, we can infer the minimum best choice of $k$ since a larger value would not affect the final connectome structure significantly. Although, the best choice was $k = 10$, we set $k = 5$ in our experiments, which means that we had larger approximation than the best choice.

We additionally demonstrate the effects of each screen method on final reconstruction error after optimization. Figure 4a shows the distribution of reconstruction error over voxels. Starting the

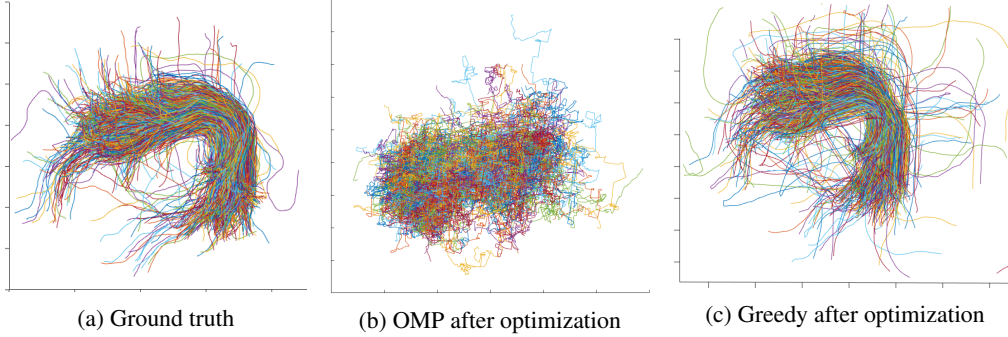

(a) Ground truth          (b) OMP after optimization          (c) Greedy after optimization

Figure 5: Solutions learned after the group sparse optimization for both screening strategies, compared to ground truth.

optimization with GreedyOrientation leads to much lower bias in the final optimization result than OMP, as demonstrated by the shift of these distributions away from the Ground Truth distribution. In Figure 4b, we show the reconstruction error on each step of optimization. The reconstruction error when initialized with orientations generated by OMP is decreasing at a rate several orders of magnitude slower than GreedyOrientation.

## 5.2 Group Sparse Optimization

After $\underline{\Phi}$ has been initialized with one of the locally greedy screening algorithms, we learn the appropriate weighting of $\underline{\Phi}$ by optimizing the global objective. We applied batch gradient decent with $15$ iterations and a dynamic step-size value which started from 1e-5 and decreased each time the algorithm could not improve the total objective error. The $\ell_1$ and group regularizer coefficients were chosen to be $10$ for most of the experiments, we tested the following values of the regularization coefficient $[10^{-3}, 10^{-2}, \ldots, 10^2, 10^3]$ and found that results were negligibly affected. For $\ell_1$ regularizer, we applied a proximal operator to truncate weights less than the threshold of $0.001$. The derivation of the gradient and optimization procedure can be found in Appendices C.2 and E.3, respectively. The visualization algorithm, for a given $\underline{\Phi}$, is given in Appendix E.4.

Figure 5 visualizes the results of $\underline{\Phi}$ after optimization with both OMP and GreedyOrientation initialization strategies. Comparing the GreedyOrientation predicted $\underline{\Phi}$ with expert $\underline{\Phi}$ shows that the group regularizer performed well in regenerating macrostructure of the Arcuate. Figure 5b shows that the OMP initialization strategy for $\underline{\Phi}$ is not appropriate for this setting, and prevents the global optimization procedure from generating the desired macrostructure.

To get a better sense for the generated fascicles, we illustrate the best and the worst fascicles for $\underline{\Phi}$ initialized with GreedyOrientation and OMP in Figure 6. GreedyOrientation produces plausible fascicles in terms of orientation, in some cases seemingly even more so than the ground truth which was obtained with a tractography algorithm. In the best case, in Figure 6a the reconstruction is highly accurate. In the worst case, in Figure 6b, GreedyOrientation produces fascicles with sharply changing direction. Looking closer, the worst reconstructed fascicles tend to be long winding fascicles with abrupt direction changes. Because the objective attempts to minimize these features during optimization, these tracts are very difficult to reconstruct. Fascicles such as these are unlikely to occur in the brain, and are likely a result of imperfect tractography methods that were used for creating the ground truth data for this experiment. Solutions with OMP are generally poor.

## 6 Conclusion and Discussion

In this work, we considered the problem of learning macroscopic brain connectomes from dMRI data. This involves inferring locations and orientations of fascicles given local measurements of diffusion of water molecules within the white-matter tissue. We proposed a new way to formulate this learning problem, using a tensor encoding. Our proposed group sparse objective facilitates the use of optimization algorithms to automatically extract brain structure, without relying on expert tractography solutions. We proposed an efficient greedy screening algorithm for this objective, and proved

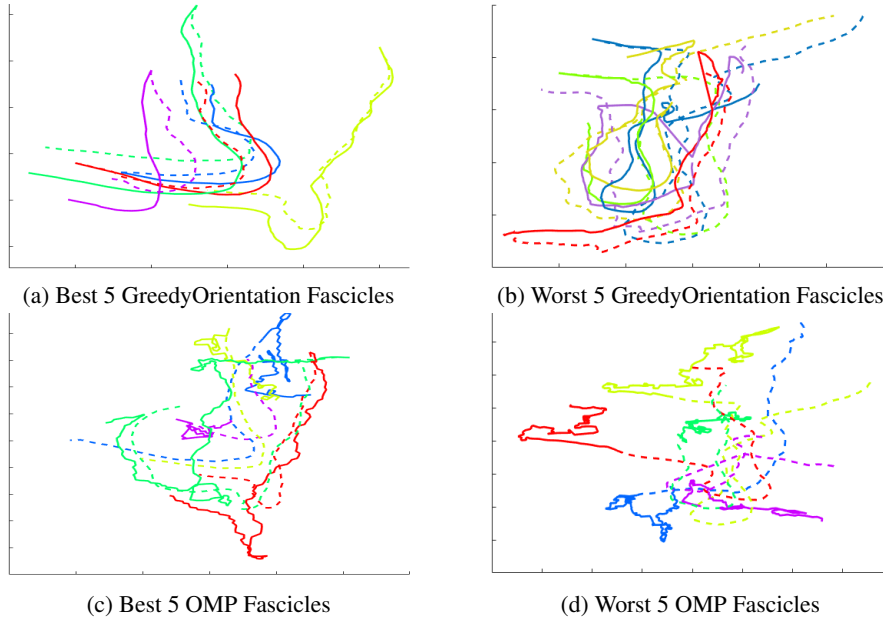

| | |
|---|---|
| (a) Best 5 GreedyOrientation Fascicles | (b) Worst 5 GreedyOrientation Fascicles |
| (c) Best 5 OMP Fascicles | (d) Worst 5 OMP Fascicles |

Figure 6: Top five best and worst fascicles for OMP and GreedyOrientation after optimization according to reconstruction error. Solid lines show the predicted $\underline{\Phi}$ and dashed lines ground truth.

approximation guarantees for the algorithm. We finally demonstrated that our specialized screening algorithm resulted in a much better orientations than a generic greedy subselection algorithm, called OMP. The solutions with our group sparse objective, in conjunction with these selected orientations, resulted in smooth fascicles and low reconstruction error of the diffusion data. We also highlighted some failures of the solution, and that more needs to be done to get fully plausible solutions.

Our tractography learning formulation has the potential to open new avenues for learning-based approaches for obtaining brain connectomes. This preliminary work was necessarily limited, focused on providing a sound formulation and providing an initial empirical investigation into the efficacy of the approximations. The next step is to demonstrate the real utility of a full tractography solution using this formulation. This will involve learning solutions across brain datasets; understanding strengths and weaknesses compared to current tractography approaches; potentially incorporating new regularizers and algorithms; and even incorporating different types of data. All of this can build on the central idea introduced in this work: using a factorization encoding to automatically learn brain structure from data.

**Acknowledgments**

This research was funded by NSERC, Amii and CIFAR. Computing was generously provided by Compute Canada and Cybera.

F.P. was supported by NSF IIS-1636893, NSF BCS-1734853, NSF AOC 1916518, NIH NCATS UL1TR002529, a Microsoft Research Award, Google Cloud Platform, and the Indiana University Areas of Emergent Research initiative "Learning: Brains, Machines, Children.

## Footnotes

[1]The original encoding uses a set of fascicles weights $\mathbf{w} \in \mathbb{R}^{N_f}$, to get $\mathbf{Y} \approx \underline{\mathbf{\Phi}} \times_1 \mathbf{D} \times_3 \mathbf{w}$. For a fixed $\underline{\mathbf{\Phi}}$, $\mathbf{w}$ was learned to adjust the magnitude of each fascicle dimension. We do not require this additional vector, because these magnitudes can be incorporated into $\underline{\mathbf{\Phi}}$ and implicitly learned when $\underline{\mathbf{\Phi}}$ is learned.

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
