[Supplementary Material]

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

## A  Maximum likelihood formulation

We first consider a maximum likelihood formulation for the reconstruction loss, paralleling the losses considered for non-matrix data. This approach involves making distributional assumptions on the matrix $\mathbf{Y}$; we begin with the standard normal, though this could be generalized to other distributions—as is commonly done for generalized linear models—without eschewing convexity. Using the matrix normal distribution [18], we can formulate the loss between $\mathbf{Y}$ and the factorization by parameterizing the matrix normal using the factorized variables. Assume $\mathbf{Y} \sim \mathcal{MN}(\mathbf{M}, \mathbf{U}, \mathbf{V})$, where $\mathcal{MN}$ is the Multivariate Normal, $\mathbf{M}$ is the mean matrix, $\mathbf{U}$ is the row variance, and $\mathbf{V}$ is the column variance. As a common simplification, we will take $\mathbf{U} = \sigma_u^2\mathbf{I}$ and $\mathbf{V} = \sigma_v^2\mathbf{I}$. Then the pdf of $\mathbf{Y}$ is

$$P(\mathbf{Y}) = \frac{\exp\left(\frac{-1}{2}\text{tr}(\mathbf{V}^{-1}(\mathbf{Y}-\mathbf{M})^\top\mathbf{U}^{-1}(\mathbf{Y}-\mathbf{M}))\right)}{(2\pi)^{N_\theta N_v/2}|\mathbf{V}|^{N_\theta/2}|\mathbf{U}|^{N_v/2}} = \frac{\exp\left(\frac{-1}{2}\sigma_v\sigma_u\text{tr}((\mathbf{Y}-\mathbf{M})^\top(\mathbf{Y}-\mathbf{M}))\right)}{(2\pi)^{N_\theta N_v/2}\sigma_v^{N_\theta^2}\sigma_u^{N_v^2}}$$

because $|\mathbf{U}| = |\sigma_u^2\mathbf{I}| = \sigma_u^{2N_v}$. This type of modeling approach assumes zero-mean, independent noise across entries in $\mathbf{Y}$, which is a common assumption [4]. Taking the negative of the log likelihood, and dropping constants which do not affect the minimum, we obtain the optimization

$$\underset{\mathbf{M}}{\text{argmin}}\ \text{tr}((\mathbf{Y}-\mathbf{M})^\top(\mathbf{Y}-\mathbf{M})) = \underset{\mathbf{M}}{\text{argmin}}\ \|\mathbf{Y}-\mathbf{M}\|_F^2$$

## B  Theoretical Properties of the Tractography Objective

An important property of this objective is that it is convex and has a unique solution for $\underline{\boldsymbol{\Phi}}$ (up to permutation), as we show in the below theorem. The convexity of the objective ensures that gradient descent can obtain optimal solutions, which is critical for both improving the objective and ensuring that accurate tractography solutions are extracted. The uniqueness is further important, because it provides an identifiable solution. For tractography, we would like to identify the fascicle structure for an individual; for an objective with multiple equivalent solutions, it is not clear which solution to select, and reflects an impreciseness in the objective. Any solution for $\underline{\boldsymbol{\Phi}}$ will always be equivalent, up to permutations of the fascicles (frontal slices, see Fig 2A - left), but should not change which fascicles are shared by which voxels.

**Theorem 1.** *The objective in* (5) *is convex in* $\underline{\boldsymbol{\Phi}}$. *Further, if the defined blocks* $\mathcal{A}$ *and* $\mathcal{V}$ *cover all possible orientations and voxels in the sense that every* $v$ *is included in at least one group* $\mathcal{G}_\mathcal{V}$ *and every orientation* $a$ *is included in at least one group* $\mathcal{G}_\mathcal{A}$, *then* (5) *has a unique solution (up to permutation).*

*Proof.* Because the sum of convex functions is convex, to show that (5) is convex, we simply need to show that each term in the objective is convex.

The first term $\|\mathbf{Y} - \underline{\boldsymbol{\Phi}} \times_1 \mathbf{D} \times_3 \mathbf{1}\|_F^2$ is convex in $\underline{\boldsymbol{\Phi}}$ because $\|\mathbf{Y} - \mathbf{M}\|_F^2$ is convex in $\mathbf{M}$, and $\mathbf{M} = \underline{\boldsymbol{\Phi}} \times_1 \mathbf{D} \times_3 \mathbf{1}$ is an affine transformation of $\underline{\boldsymbol{\Phi}}$. The composition of an affine function and a convex function is convex.

The second term is the sum of several functions of $\underline{\boldsymbol{\Phi}}$, which only consider subparts of $\underline{\boldsymbol{\Phi}}$. If each of these functions in the group regularizer is convex, then the regularizer is composed of the sum of convex functions and so is itself convex. Let $R_{\mathcal{G}_\mathcal{A},\mathcal{G}_\mathcal{V},f}(\underline{\boldsymbol{\Phi}}) = \|\mathbf{x}_{\mathcal{G}_\mathcal{A},\mathcal{G}_\mathcal{V},f}\|_2$. This function only changes when elements in $\underline{\boldsymbol{\Phi}}$ related to $\mathcal{G}_\mathcal{A}, \mathcal{G}_\mathcal{V}, f$ change, and is otherwise constant. However, since a constant function is convex, $R_{\mathcal{G}_\mathcal{A},\mathcal{G}_\mathcal{V},f}$ is convex in the entries of $\underline{\boldsymbol{\Phi}}$ that are ignored. Let $\underline{\boldsymbol{\Phi}}_{\mathcal{G}_\mathcal{A},\mathcal{G}_\mathcal{V},f}$ be the entries in $\underline{\boldsymbol{\Phi}}$ that give $\mathbf{x}_{\mathcal{G}_\mathcal{A},\mathcal{G}_\mathcal{V},f} = [\|\underline{\boldsymbol{\Phi}}_{\mathcal{G}_\mathcal{A},v_1,f}\|_1, \ldots, \|\underline{\boldsymbol{\Phi}}_{\mathcal{G}_\mathcal{A},v_n,f}\|_1]$ for $v_i \in \mathcal{G}_\mathcal{V}$. We can consider $\|\mathbf{x}_{\mathcal{G}_\mathcal{A},\mathcal{G}_\mathcal{V},f}\|_2$ as a vector composition, of $g : \mathbb{R}^k \to \mathbb{R}^n$ and $h : \mathbb{R}^n \to \mathbb{R}$, where $g(\underline{\boldsymbol{\Phi}}_{\mathcal{G}_\mathcal{A},\mathcal{G}_\mathcal{V},f}) = \mathbf{x}_{\mathcal{G}_\mathcal{A},\mathcal{G}_\mathcal{V},f}$ and $h(\mathbf{x}) = \|\mathbf{x}\|_2$. The resulting composition is $h(g(\underline{\boldsymbol{\Phi}}_{\mathcal{G}_\mathcal{A},\mathcal{G}_\mathcal{V},f})) = \|\mathbf{x}_{\mathcal{G}_\mathcal{A},\mathcal{G}_\mathcal{V},f}\|_2$. Each $g_i$ of the vector-valued function $g$ is convex in $\underline{\boldsymbol{\Phi}}_{\mathcal{G}_\mathcal{A},\mathcal{G}_\mathcal{V},f}$ because it applies an $\ell_1$ norm—which is convex—on a subset of $\underline{\boldsymbol{\Phi}}_{\mathcal{G}_\mathcal{A},\mathcal{G}_\mathcal{V},f}$. Further, $h$ is convex in $\mathbf{x}$, and non-decreasing in each $\mathbf{x}_i = g_i(\underline{\boldsymbol{\Phi}}_{\mathcal{G}_\mathcal{A},\mathcal{G}_\mathcal{V},f})$, because $h$ is a norm. Therefore, the composition $h(g(\cdot))$ is convex. Therefore, because $R_{\mathcal{G}_\mathcal{A},\mathcal{G}_\mathcal{V},f}$ is convex w.r.t. $\underline{\boldsymbol{\Phi}}_{\mathcal{G}_\mathcal{A},\mathcal{G}_\mathcal{V},f}$ and constant w.r.t. all other values in $\underline{\boldsymbol{\Phi}}$, we know that $R_{\mathcal{G}_\mathcal{A},\mathcal{G}_\mathcal{V},f}$ is convex in $\underline{\boldsymbol{\Phi}}$. Since $R(\underline{\boldsymbol{\Phi}}) = \sum_{f \in \mathcal{F}} \sum_{\mathcal{G}_\mathcal{V} \in \mathcal{V}} \sum_{\mathcal{G}_\mathcal{A} \in \mathcal{A}} R_{\mathcal{G}_\mathcal{A},\mathcal{G}_\mathcal{V},f}(\underline{\boldsymbol{\Phi}})$ is a sum of convex function, it is convex.

To show uniqueness, we need to show that the regularizer is strictly convex. Because the sum of a strictly convex and convex function is again strictly convex, the resulting objective is itself strictly convex and so must have a unique minimum. Each component of the regularizer only considers a subset of $\underline{\mathbf{\Phi}}$; however, as long as each possible entry in $\underline{\mathbf{\Phi}}$ is considered at least once in one of these blocks, then that component of $\underline{\mathbf{\Phi}}$ has a strictly convex regularizer on it in the objective, because norms are strictly convex. $\qquad\square$

## C   Sparse tensor factorization algorithm

In this section, we develop an algorithm to optimize our extremely sparse, high-dimensional objective. A common strategy for sparse optimization problems is to first perform screening on the coefficients— which corresponds to all of $\underline{\mathbf{\Phi}}$ in this setting—to avoid modifying coefficients that will remain zero. A number of generalized screening approaches have been developed for general sparse problems, either with a static screening before the start of the optimization [19] or with a dynamic screening that adjust the set of feasible coefficients during the optimization [6]. We propose a specialized static screening, where we first select a set of feasible orientations for each voxel. This static screening on the entries in $\underline{\mathbf{\Phi}}$ significantly reduces the cost per iteration of gradient descent and reduces the number of iterations. This significantly speeds up the optimization, without incurring much approximation error, because of the approximation guarantees of the static screening approach. We first highlight why standard matrix and tensor factorization algorithms are not suitable for this problem, and then derive our specialized solver.

### C.1   Issues with using standard matrix or tensor factorization algorithms

A natural community to turn to for solutions to obtain $\underline{\mathbf{\Phi}}$ is tensor factorization. The goal in this work is to factorize a matrix $\mathbf{Y}$ in a tensor $\underline{\mathbf{\Phi}}$, for a given dictionary $\mathbf{D}$, such that $\mathbf{Y} = \underline{\mathbf{\Phi}} \times_1 \mathbf{D} \times_3 \mathbf{1}$. Much of the work in tensor factorization, however, has focused on decomposing tensors into a set of matrices, with a small core tensor with the Tucker decomposition focused on low rank tensor factorizations (see [10] for a thorough overview). A few of these works have examined how to obtain a sparse core tensor, but towards the aim of either enforcing uniqueness [16] or to obtain core tensors that are more efficient to store and use [34, 7, 25, 11]. There has been some work on factorizing large sparse tensors, for tensor-SVD [1, 33]; again, however, their goal is to factorize a sparse tensor, which differs from our goal to factorize a dense matrix into an extremely sparse tensor with a particular structure. The most closely related algorithm is derived for low-rank regularizers for non-negative tensor factorization [32], but it is not designed for large sparse tensors.

### C.2   Computing the subgradient of the objective

Once these orientations are set per voxel, we can much more efficiently compute the gradient for the objective, because the sum over groups significantly reduces,

$$\sum_{\mathcal{G}_{\mathcal{V}} \in \mathcal{V}} \sum_{\mathcal{G}_{\mathcal{A}} \in \mathcal{A}} \|\mathbf{x}_{\mathcal{G}_{\mathcal{A}}, \mathcal{G}_{\mathcal{V}}, f}\|_2 = \sum_{\mathcal{G}_{\mathcal{V}} \in \mathcal{V}} \sum_{\mathcal{G}_{\mathcal{A}} \in \mathcal{A}(\mathcal{G}_{\mathcal{V}})} \|\mathbf{x}_{\mathcal{G}_{\mathcal{A}}, \mathcal{G}_{\mathcal{V}}, f}\|_2$$

where $\mathcal{A}(\mathcal{G}_{\mathcal{V}}) = \{\mathcal{G}_{\mathcal{A}} \in \mathcal{A} \mid \mathcal{G}_{\mathcal{A}} \cap S(v) \neq \emptyset, v \in \mathcal{G}_{\mathcal{V}}\}$. The set $\mathcal{A}(\mathcal{G}_{\mathcal{V}})$ only includes groups with orientations that are active for at least one voxel in $\mathcal{G}_{\mathcal{V}}$.

Let there be $N_{G_a}$ atom groups and $N_{G_v}$ voxel groups. We use $\mathbf{G}_A \in \{0,1\}^{N_a \times N_{G_a}}$ and $\mathbf{G}_V \in \{0,1\}^{N_v \times N_{G_v}}$ to denote if an atom or a voxel belongs to a group. Specifically, if $\mathbf{G}_A(a,g) = 1$, then atom $a$ belongs to group $g$; otherwise it does not. The subderivative of the group regularizer w.r.t. $\underline{\mathbf{\Phi}}(a,v,f)$ is

$$\lambda_g \sum_{a_g}^{N_{G_a}} \mathbf{G}_A(a, a_g) \sum_{v_g}^{N_{G_v}} \mathbf{G}_V(v, v_g) \frac{\sum\limits_{a_i \in \mathcal{G}_{a_g}} |\underline{\mathbf{\Phi}}(a_i, v, f)|}{\underline{\mathbf{A}}(a_g, v_g, f)} \operatorname{sign}(\underline{\mathbf{\Phi}}(a, v, f))$$

where $\underline{\mathbf{A}} = \sqrt{\left(|\underline{\boldsymbol{\Phi}}| \times_1 \mathbf{G}_A^\top\right)^2 \times_2 \mathbf{G}_V^\top}$. Additionally, we include a standard $\ell_1$ regularizer on all of $\underline{\boldsymbol{\Phi}}$ to further promote sparsity. The full subgradient of the objective w.r.t. $\underline{\boldsymbol{\Phi}}$ is

$$
\begin{aligned}
\nabla_{\underline{\boldsymbol{\Phi}}} =& \mathbf{D}^\top (\underline{\boldsymbol{\Phi}} \times_1 \mathbf{D} \times_3 \mathbf{1} - \mathbf{Y})\mathbf{1}^\top \\
&+ \lambda_g \left( \left\{ \left(|\underline{\boldsymbol{\Phi}}| \times_1 \mathbf{G}_A^\top\right) \circ \left( \frac{1}{\underline{\mathbf{A}}} \times_2 \mathbf{G}_V \right) \right\} \times_1 \mathbf{G}_A \right) \circ \operatorname{sign}(\underline{\boldsymbol{\Phi}}) \\
&+ \lambda_1 \operatorname{sign}(\underline{\boldsymbol{\Phi}})
\end{aligned}
\tag{6}
$$

On each step, we step in the direction of the negative of the gradient, with a fixed stepsize which we set to $\eta = 1e - 3$, until the objective improvement is below a threshold $1e - 4$ or until a maximum number of iterations is reached. In addition to the initial screening, we can obtain some speed improvements on each step by only computing the gradient for currently active elements in $\underline{\boldsymbol{\Phi}}$. For any zeroed elements in $\underline{\boldsymbol{\Phi}}$, the gradient of the regularizer would be zero, as the regularizer prefers each element be zero.

### C.3 Derivation of Forward Selection for Orientations

To derive an efficient forward selection for the orientations, we need to make each greedy step efficient. For a fixed voxel $v$, the greedy algorithm selects the atom $a$ which most increases $\bar{g}$:

$$
\max_{a \notin S} \bar{g}(S \cup \{a\}) = \max_{a \notin S} g(S \cup \{a\}) + \sum_{s \in S} g(\{s\}) + g(\{a\}).
$$

To compute $g(\{a\})$, we can compute this upfront for each $a$ and store it before doing the full greedy optimization. For the greedy optimization, the most naive solution would be to compute the full regression solution for each new subset $S \cup \{a\}$, to obtain $g(S \cup \{a\})$. Unfortunately, this brute-force approach is too expensive. Because of the structure of $\bar{g}$, however, we can take advantage of the solution on the previous step, to compute the solution on this step.

We provide the recursive update mechanism in Lemma 1. Let $\mathbf{y} \in \mathbb{R}^{N_\theta}$ be the diffusion information for one voxel. For given orientations $S$ with $|S| = k$, let $\mathbf{D}_S \in \mathbb{R}^{N_\theta \times k}$ be the subset of columns in $\mathbf{D}$ corresponding to orientations in $S$. Using similar subscript notation, with $a \notin S$ a new atom not yet chosen in $S$, let

$$
\begin{aligned}
\mathbf{C}_S &= \mathbf{D}_S^\top \mathbf{D}_S \\
\mathbf{b}_S &= \mathbf{D}_S^\top \mathbf{y} \\
C_a &= \mathbf{D}_a^\top \mathbf{D}_a \\
b_a &= \mathbf{D}_a^\top \mathbf{y} \\
\mathbf{c}_{S,a} &= \mathbf{D}_S^\top \mathbf{D}_a
\end{aligned}
$$

The squared multiple correlation is

$$
g(S) = \mathbf{b}_S^\top \mathbf{C}_S^{-1} \mathbf{b}_S
$$

and $g(\{a\}) = C_a^{-1} b_a^2$. We provide the following lemma to obtain an efficient mechanism to compute $g(S \cup \{a\})$ for each $a$. These recursive updates are similar to the updates given by [24], for OMP.

**Lemma 1.** *Given* $\mathbf{C}_S^{-1} \in \mathbb{R}^{k \times k}$, $\mathbf{b}_S$ *and* $g(S)$, *for*

$$
\begin{aligned}
\mathbf{c} &= \mathbf{C}_S^{-1} \mathbf{c}_{S,a} \\
\nu &= (C_a - \mathbf{c}_{S,a}^\top \mathbf{c})^{-1}
\end{aligned}
$$

*we get that*

$$
g(S \cup \{a\}) = g(S) + \nu(\mathbf{b}_S^\top \mathbf{c} - b_a)^2
$$

*Further*

$$
\bar{g}(S \cup \{a\}) = \bar{g}(S) + \nu(\mathbf{b}_S^\top \mathbf{c} - b_a)^2 + C_a^{-1} b_a^2
$$

*Proof.* We know that $g(S \cup \{a\}) = \mathbf{b}_{S\cup\{a\}}^\top \mathbf{C}_{S\cup\{a\}}^{-1} \mathbf{b}_{S\cup\{a\}}$. We need to compute the inverse of $\mathbf{C}_{S\cup\{a\}}$ using the inverse of $\mathbf{C}_S$. We use the general block matrix inversion formula

$$\mathbf{C}_{S\cup\{a\}}^{-1} = \left[ \begin{array}{cc} \mathbf{C}_S & \mathbf{c}_{S,a} \\ \mathbf{c}_{S,a}^\top & C_a \end{array} \right]^{-1}$$

$$= \left[ \begin{array}{cc} \mathbf{C}_S^{-1} + \nu \mathbf{C}_S^{-1} \mathbf{c}_{S,a} \mathbf{c}_{S,a}^\top \mathbf{C}_S^{-1} & -\nu \mathbf{C}_S^{-1} \mathbf{c}_{S,a} \\ -\nu \mathbf{c}_{S,a}^\top \mathbf{C}_S^{-1} & \nu \end{array} \right]$$

Therefore,

$$g(S \cup \{a\}) = \mathbf{b}_{S\cup\{a\}}^\top \mathbf{C}_{S\cup\{a\}}^{-1} \mathbf{b}_{S\cup\{a\}}$$

$$= \mathbf{b}_S^\top \mathbf{C}_S^{-1} \mathbf{b}_S + \nu \mathbf{b}_S^\top \mathbf{C}_S^{-1} \mathbf{c}_{S,a} \mathbf{c}_{S,a}^\top \mathbf{C}_S^{-1} \mathbf{b}_S - 2\nu \mathbf{b}_S^\top \mathbf{C}_S^{-1} \mathbf{c}_{S,a} b_a + \nu b_a^2$$

$$= g(S) + \nu (\mathbf{b}_S^\top \mathbf{c})^2 - 2\nu b_a \mathbf{b}_S^\top \mathbf{c} + \nu b_a^2$$

$$= g(S) + \nu (\mathbf{b}_S^\top \mathbf{c} - \nu b_a)$$

Using this, we can see that

$$\bar{g}(S \cup \{a\}) = g(S \cup \{a\}) + \sum_{s \in S \cup \{a\}} g(\{s\})$$

$$= g(S) + \nu (\mathbf{b}_S^\top \mathbf{c} - \nu b_a)^2 + \sum_{s \in S} g(\{s\}) + g(\{a\})$$

$$= \bar{g}(S) + \nu (\mathbf{b}_S^\top \mathbf{c} - \nu b_a)^2 + g(\{a\})$$

completing the proof, because $g(\{a\}) = C_a^{-1} b_a^2$. □

Given this result, the computation of $g(S \cup \{a\})$ for one atom $a$ costs $O(kN_\theta + k^2) = O(kN_\theta)$, since $N_\theta > k$. To compute $g$ for each $a$, therefore, costs a total of $O(kN_\theta N_a)$. We summarize the greedy algorithm for computing the directions for a voxel in Algorithm 1. A new point is added to greedily maximize $\bar{g}(S)$, until $S$ has $k$ directions.

---

**Algorithm 1** GreedyOrientation: greedy algorithm to select orientations for each voxel

---

1: Input dictionary $\mathbf{D}$, maximum number of orientations $k$, diffusion signal $\mathbf{y}$
2: // Compute $g(\{a\})$ for each $a$, $\mathbf{g} = \text{diag}(\mathbf{D}^\top \mathbf{D})^{-1}(\mathbf{D}^\top \mathbf{y})^2$
3: $\mathbf{c} \leftarrow \mathbf{D}^\top \mathbf{D}$
4: $\mathbf{b} \leftarrow \mathbf{D}^\top \mathbf{y}$
5: $\mathbf{g} \leftarrow \mathbf{0}$
6: $a_{\max} \leftarrow -1, g_{\max} \leftarrow 0$
7: **for** $a = 1, \ldots, N_a$ **do**               ▷ $O(N_\theta N_a)$
8:     $\mathbf{g}(a) \leftarrow (\mathbf{b}(a))^2 / \mathbf{c}(a, a)$
9:     **if** $g_{\max} < \mathbf{g}(a)$ **then**
10:         $g_{\max} \leftarrow \mathbf{g}(a), a_{\max} \leftarrow a$
11: $S \leftarrow a_{\max}$
12: $\mathbf{C}^{-1} \leftarrow 1/\mathbf{c}(a_{\max}, a_{\max})$
13: **for** $i = 2, \ldots, k$ **do**
14:     // Compute $g(S \cup \{a\})$ for every $a$
15:     $\mathbf{g}_S, \nu \leftarrow \text{ComputeGain}(S, \mathbf{C}^{-1}, \mathbf{c}, \mathbf{b})$        ▷ $O(kN_\theta)$
16:     $\bar{\mathbf{g}} \leftarrow \mathbf{g}_S + \mathbf{g}$               ▷ $\bar{\mathbf{g}} \in \mathbb{R}_a^N$
17:     $a_{\max} \leftarrow \arg\max_{a \notin S} \bar{\mathbf{g}}(a)$         ▷ $O(N_a)$
18:     $\mathbf{C}^{-1} \leftarrow \left[ \begin{array}{cc} \mathbf{C}^{-1} + \nu(a_{\max})\mathbf{C}^{-1}\mathbf{c}(S, a_{\max})\mathbf{c}(S, a_{\max})^\top \mathbf{C}^{-1} & -\nu(a_{\max})\mathbf{C}^{-1}\mathbf{c}(S, a_{\max}) \\ -\nu(a_{\max})\mathbf{c}_{S,a}^\top \mathbf{C}_S^{-1} & \nu(a_{\max}) \end{array} \right]$
19:     $S \leftarrow S \cup \{a_{\max}\}$
20: **Output:** $S$

---

---

**Algorithm 2** ComputeGain$(S, \mathbf{C}^{-1}, \mathbf{c}, \mathbf{b})$

---

1: **for** $a = 1, \ldots, N_a$ **do**
2: $\quad \tilde{\mathbf{c}} \leftarrow \mathbf{C}^{-1}\mathbf{c}(S, a)$
3: $\quad \nu(a) \leftarrow (\mathbf{c}(a,a) - \mathbf{c}(S,a)^\top \tilde{\mathbf{c}})^{-1}$
4: $\quad \mathbf{g}_S(a) \leftarrow \nu(a)(\mathbf{b}(S)^\top \tilde{\mathbf{c}} - \mathbf{b}(a))^2$
5: **Output:** $\mathbf{g}_S, \nu$

---

## D   Theoretical guarantees of the greedy screening strategy

We can obtain approximation guarantees from the fact that the approximately submodular function for GreedyOrientation has at least as good a submodularity ratio as the typical forward selection function $g$. The submodularity ratio is defined as

$$\gamma(g) \overset{\text{def}}{=} \min_{S, L \in \mathcal{S}, S \cap L = \emptyset} \frac{\sum_{y \in S} g(L \cup \{y\}) - g(L)}{g(L \cup S) - g(L)} \tag{7}$$

for non-negative functions $g : \mathcal{P}(\mathcal{S}) \to \mathbb{R}^+$. If $g$ is a monotone function and $\gamma(g) \geq 1$, then $g$ is submodular. Otherwise, for $\gamma(g) < 1$, the function is not submodular and is instead called approximately submodular for $\gamma(g)$ close to 1. The closer $\gamma(g)$ is to 1, the better the approximation guarantees of greedy algorithms on these functions, with the best approximation guarantees for $\gamma(g) \geq 1$.

In the following theorem, we show that our GreedyOrientation function $\bar{g}$ has a submodularity ratio that is no worse than ForwardSelection. The proof highlights that in fact the ratio is likely strictly better.

**Theorem 2.** *For $g : \mathcal{P}(\mathcal{S}) \to \mathbb{R}^+$ a monotone function, and*

$$\bar{g}(S) \overset{\text{def}}{=} g(S) + \sum_{s \in S} g(\{s\})$$

*then*

$$\gamma(\bar{g}) \geq \gamma(g).$$

*Proof.* For clarity, we introduce notation for the numerator and denominator of the submodularity ratio:

$$\gamma_N(g, L, S) \overset{\text{def}}{=} \sum_{y \in S} g(L \cup \{y\}) - g(L)$$

$$\gamma_D(g, L, S) \overset{\text{def}}{=} g(L \cup S) - g(L)$$

Notice that

$$\bar{g}(L) = g(L) + \sum_{x \in L} g(\{x\})$$

$$\bar{g}(L \cup \{y\}) = g(L) + \sum_{x \in L} g(\{x\}) + g(\{y\})$$

$$\bar{g}(L \cup S) = g(L \cup S) + \sum_{x \in L} g(\{x\}) + \sum_{y \in S} g(\{y\})$$

giving

$$\begin{aligned}
\gamma_N(\bar{g}, L, S) &= \sum_{y \in S} \bar{g}(L \cup \{y\}) - \bar{g}(L) \\
&= \sum_{y \in S} g(L \cup \{y\}) + g(\{y\}) - g(L) \\
&= \gamma_N(g, L, S) + \sum_{y \in S} g(\{y\})
\end{aligned}$$

and

$$\gamma_D(\bar{g}, L, S) = \bar{g}(L \cup S) - \bar{g}(L)$$
$$= g(L \cup S) + \sum_{y \in S} g(\{y\}) - g(L)$$
$$= \gamma_D(g, L, S) + \sum_{y \in S} g(\{y\}).$$

If we let $a_S \stackrel{\text{def}}{=} \sum_{y \in S} g(\{y\}) \geq 0$, then we get that

$$\gamma(\bar{g}) = \min_{S, L \in \mathcal{S}, S \cap L = \emptyset} \frac{\gamma_N(\bar{g}, L, S)}{\gamma_D(\bar{g}, L, S)}$$
$$= \min_{S, L \in \mathcal{S}, S \cap L = \emptyset} \frac{\gamma_N(g, L, S) + a_S}{\gamma_D(g, L, S) + a_S}$$
$$\geq \min_{S, L \in \mathcal{S}, S \cap L = \emptyset} \frac{\gamma_N(g, L, S)}{\gamma_D(g, L, S)} \qquad \triangleright \text{ because } a_S > 0$$
$$= \gamma(g)$$

completing the proof.

$\square$

The following result now easily follows, from Theorem 4.2 [13] for approximately submodular functions.

**Corollary 1.** *The set of orientations $S$ chosen by GreedyOrientation satisfies*

$$\bar{g}(S) \geq \left(1 - e^{-\gamma(\bar{g})}\right) OPT$$

*where $OPT = \bar{g}(S^*)$ for the optimal selection $S^*$ such that $|S^*| = k$.*

## E    Experiments and Results

### E.1    Computational Resources

All experiments in this paper were run using an Intel Xeon processor from 2014 with 8 cores at 2.4Ghz each and with 32GB of ram. The code relies heavily on the sparsity of the data, using efficient sparse tensor operations to minimize memory usage and necessary computational resources. Scaling up to larger dimensions or using higher resolution data would greatly increase the total number of entries in the tensors (including empty values), but would increase the number of *active* entries at a much lower rate.

### E.2    Generating data

To generate synthetic data for our experiments, we used the dMRI data of one subject's brain and applied an expert tractography algorithm to over-generate fascicles, $N_f = 500,000$, as our candidate connectomes to fit the LiFE model [21]. LiFE takes any connectome as input and predicts demeaned diffusion measurements as output in order to evaluate tractography algorithms. We employed LiFE to purify connectomes and prune the number of candidate fascicles. It zeros out the weights of fascicles that do not have significant contribution in reconstructing diffusion signal. This reduces the size of fascicle set by a factor of $5$ and decreases the reconstruction error of diffusion signal compared to the tractography one. We also applied ENCODE to unified encoding of the brain structure and dMRI signal by applying dictionary $D$. [2] The ENCODE model generated the predicted signal using the two major structures that we consider in this paper, the Arcuate Fasciculus and the ARC-SLF; which is the Arcuate combined with SLF1.

## E.3  Optimization algorithm for mapping connectomes of the brain

---

**Algorithm 3** Brain: Mapping Brain Connectomes

---

**Input:** dMRI signal $\mathbf{Y}$, expert three dimensional sparse tensor $\underline{\mathbf{\Phi}}_e$, dictionary $\mathbf{D}$, weights or fascicles' contribution $\mathbf{w}$, voxels vicinity $\mathbf{Vv}$ matrix (group information $\mathcal{G}_\mathcal{V}$) and atoms vicinity $\mathbf{Av}$ matrix (group information $\mathcal{G}_\mathcal{A}$), maximum iteration *max_iter*, step size *step_size*, termination condition *tolerance*

**Ensure:** $\|\mathbf{Y} - \underline{\mathbf{\Phi}} \times_1 \mathbf{D} \times_3 \mathbf{1}\|_F^2 + \lambda \sum_{f \in \mathcal{F}} \sum_{\mathcal{G}_\mathcal{V} \in \mathcal{V}} \sum_{\mathcal{G}_\mathcal{A} \in \mathcal{A}} \|\underline{\mathbf{\Phi}}_{\mathcal{G}_\mathcal{A}, \mathcal{G}_\mathcal{V}, f}\|_2$ is minimum

1: **for** $f = 1, \dots, N_f$ **do**
2:     $\underline{\mathbf{\Phi}}_e(:,:,f) = \underline{\mathbf{\Phi}}_e(:,:,f) * \mathbf{w}(f)$            ▷ Fold $\mathbf{w}$ in $\underline{\mathbf{\Phi}}_e$
3: voxels ← non-zero($\underline{\mathbf{\Phi}}_e$, 2) ▷ Find necessary voxels from $\underline{\mathbf{\Phi}}_e$ (Non-zero elements after summing up the other two dimensions)
4: atoms ← non-zero($\underline{\mathbf{\Phi}}_e$, 1)   ▷ Find necessary atoms(orientations) from $\underline{\mathbf{\Phi}}_e$ (Non-zero elements after summing up the other two dimensions)
5: $\mathbf{Y} = \mathbf{Y}(:, \text{voxels})$                      ▷ Remove unnecessary voxels of $\mathbf{Y}$
6: fascicles ← fascicles(non-zero(w)) ▷ Remove unnecessary fascicles where contribution (weight) is 0
7: $N_a, N_v, N_f \leftarrow$ size(atoms), size(voxels), size(fascicles)
8: **for** $v = 1, \dots, N_v$ **do**
9:     $\mathbf{aA} \leftarrow$ GreedyOrientation($\mathbf{D}, k$)       ▷ Find indices of active atoms $\mathbf{aF}$ with Algorithm 1
10:    $\mathbf{aF} \leftarrow$ non-zero($\underline{\mathbf{\Phi}}_e$, 3)             ▷ Find indices of active fascicles $\mathbf{aF}$ from $\underline{\mathbf{\Phi}}_e$
11:    $\underline{\mathbf{\Phi}}(\mathbf{aA}, v, \mathbf{aF}) \leftarrow Initialization()$   ▷ Initialize $\underline{\mathbf{\Phi}}$ with non-zero values where atoms and fascicles are active
12: $\mathbf{G}_V \leftarrow$ find($\mathbf{Vv}$)  ▷ $\mathbf{G}_V(i,j) \in \{0,1\}$ Denotes that if voxel i is in the neighborhood of voxel j
13: $\mathbf{G}_A \leftarrow$ find($\mathbf{Av}$)    ▷ $\mathbf{G}_A(i,j) \in \{0,1\}$ Denotes that if atom i is in the neighborhood of atom j
14: $\mathbf{Emask} \leftarrow (\underline{\mathbf{\Phi}} \times_2 \mathbf{G}_V) \times_1 \mathbf{G}_A$                  ▷ Entry mask tensor
15: $\mathbf{Fmask} \leftarrow \mathbf{Emask} \times_1 \mathbf{1}$                      ▷ Fascicles Mask matrix
16: $\mathbf{Amask} \leftarrow \mathbf{Emask} \times_3 \mathbf{1}$                      ▷ Atoms Mask matrix
17: $\mathbf{Fscreen} \leftarrow \underline{\mathbf{\Phi}} \times_1 \mathbf{1}$     ▷ Fascicles Screen matrix. Unlike $\mathbf{Fmask}$, this screen matrix does not contain group information
18: $\mathbf{Ascreen} \leftarrow \underline{\mathbf{\Phi}} \times_3 \mathbf{1}$     ▷ Atoms Screen matrix. Unlike $\mathbf{Amask}$, this screen matrix does not contain group information
19: $\mathbf{Y_{diff}} \leftarrow \mathbf{Y} - \underline{\mathbf{\Phi}} \times_1 \mathbf{D} \times_3 \mathbf{1}$
20: $R(\underline{\mathbf{\Phi}}) \leftarrow \sum_{f \in \mathcal{F}} \sum_{\mathcal{G}_\mathcal{V} \in \mathcal{V}} \sum_{\mathcal{G}_\mathcal{A} \in \mathcal{A}} \|\underline{\mathbf{\Phi}}_{\mathcal{G}_\mathcal{A}, \mathcal{G}_\mathcal{V}, f}\|_2$
21: lnew ← $\|\mathbf{Y_{diff}}\|_F^2 + \lambda R(\underline{\mathbf{\Phi}})$
22: niter ← 1
23: **repeat**
24:     lold ← lnew
25:     grad_p1_x ← $\mathbf{D}^\top (\underline{\mathbf{\Phi}} \times_1 \mathbf{D} \times_3 \mathbf{1} - \mathbf{Y}) \mathbf{1}^\top$
26:     grad_g1_x1 ← $|\underline{\mathbf{\Phi}}| \times_1 \mathbf{G}_A^\top$    ▷ O(number of nonzero elements in $\underline{\mathbf{\Phi}}\times$ number of nonzero elements in $\mathbf{G}_A$)
27:     $\underline{\mathbf{A}} \leftarrow \sqrt{\text{grad\_g1\_x1}^2 \times_2 \mathbf{G}_V^\top}$
28:     grad_g1_x3 ← $\frac{1}{\underline{\mathbf{A}}} \times_2 \mathbf{G}_V$                   ▷ O($N_v \times$ number of nonzero elements in $\underline{\mathbf{A}}$)
29:     grad_g1_x4 ← grad_g1_x3 ∘ grad_g1_x1
30:     grad_g1_v ← (grad_g1_x4 $\times_1 \mathbf{G}_A$) ∘ sign($\underline{\mathbf{\Phi}}$)
31:     g ← grad_p1_x $+\lambda_g$ grad_g1_v $+ \lambda_1$ sign($\underline{\mathbf{\Phi}}$)
32:     Mask or Screen elements in g with $\mathbf{Fmask}, \mathbf{Amask}$ or $\mathbf{Fscreen}, \mathbf{Ascreen}$
33:     $\underline{\mathbf{\Phi}} \leftarrow \underline{\mathbf{\Phi}} - $ step_size $* g$
34:     $\mathbf{Y_{diff}} \leftarrow \mathbf{Y} - \underline{\mathbf{\Phi}} \times_1 \mathbf{D} \times_3 \mathbf{1}$
35:     $R(\underline{\mathbf{\Phi}}) \leftarrow \sum_{f \in \mathcal{F}} \sum_{\mathcal{G}_\mathcal{V} \in \mathcal{V}} \sum_{\mathcal{G}_\mathcal{A} \in \mathcal{A}} \|\underline{\mathbf{\Phi}}_{\mathcal{G}_\mathcal{A}, \mathcal{G}_\mathcal{V}, f}\|_2$
36:     lnew ← $\|\mathbf{Y_{diff}}\|_F^2 + \lambda R(\underline{\mathbf{\Phi}})$
37:     niter ← niter + 1
38: **until** $lold - lnew < lold * tolerance || niter > max\_iter$
39: set small values in $\underline{\mathbf{\Phi}}$ zeros
40: **Output:** $\underline{\mathbf{\Phi}}$

---

Figure 7: Results using the larger **ARC-SLF** dataset. **(a)**: Number of missing orientations in the candidate set generated by each screening algorithm, averaged over all voxels. **(b)**: Angular distance between orientations in the candidate set and the ground truth by voxel. **(c)**: Average of the possible minimum angular distance per voxel given some linear combination of orientations in candidate sets versus the ground truth.

## E.4 Visualization algorithm

This section explains the visualization algorithm of the brain connectomes based on the sparse tensor $\underline{\mathbf{\Phi}}$. Each entry of this sparse tensor represents one node, each having an orientation, being located in a voxel, and belonging to a fascicle. There is no structure in Phi to indicate what order to put nodes in. The nodes do not contain any positional information of the space and this leads to an ambiguity in the accuracy of their order. The only positional information in hand is the coordinates of the voxels. Therefore, displaying an accurate $\underline{\mathbf{\Phi}}$ is itself a challenging problem due to many possible permutations of nodes in a voxel for each fascicle.

Our goal is to go over fascicles one by one and try to plot each at a time. We approach this by selecting one voxel that the fascicle passes through and has the fewest number of neighbouring voxels which also containing the same fascicle. A voxel with these properties should be at one end of the fascicle. Then the algorithm examines all surrounding voxels and chooses the pair of nodes with the smallest angular distance between them with one in the first voxel and one in the neighbours. We plot the nodes in the first voxel so that the last node in that voxel is the one chosen. Then we move forward through each voxel plotting first the node chosen in the previous pairing followed by the rest of the nodes in the voxel greedily chosen based on angular distance from the last plotted node.

## E.5 Angular distance evaluation measurement

The goal here is to measure a more precise metric for the angular differences of the nodes in $\underline{\mathbf{\Phi}}$-predict and $\underline{\mathbf{\Phi}}$-expert. It is not a trivial task to measure this metric since a more precise measurement requires finding a one-by-one relationship between the nodes in $\underline{\mathbf{\Phi}}$-predict and $\underline{\mathbf{\Phi}}$-expert. A reasonable way of doing that is to loop over each individual orientation per fascicle-voxel in $\underline{\mathbf{\Phi}}$-expert, $a_{exp} \in \underline{\mathbf{\Phi}}_{exp}(:,v,f)$, and find the optimal solution of active orientations from the candidate set corresponding to the same voxel in $\underline{\mathbf{\Phi}}$-predict so that they could better regenerate the diffusion signal of $\underline{\mathbf{\Phi}}_{exp}(a_{exp},v,f) \times \mathbf{D}(:,a_{exp})$. Then the angular distance of the vector-sum of activated nodes in $\underline{\mathbf{\Phi}}$-predict with $a_{exp}$ would be calculated and the average angular difference over all $a_{exp}$ per voxel would be reported.

## E.6 Evaluation results on ARC-SLF

In this section, we demonstrate the evaluation results on ARC-SLF. This tract has $N_a = 1057$, $N_v = 15033$, $N_f = 1100$, and $N_\theta = 96$.

**Algorithm 4** Visualize $\mathbf{\Phi}$, the structure of connectomes

**Input:** Any three dimensional sparse tensor $\mathbf{\Phi}$, matrix $\mathbf{A}$ to map the indices of atoms in $\mathbf{\Phi}$ to the Cartesian components of the direction vector of that atom, and matrix $\mathbf{V}$ to map the indices of voxels in $\mathbf{\Phi}$ to the Cartesian coordinates of that voxel

1: **for** $f = 1, \ldots, N_f$ **do**
2:      seen_v $\leftarrow \emptyset$    $\triangleright$ Initialize set to keep track of which voxels have been visited for fascicle $f$
3:      $\mathbf{v} \leftarrow$ get_voxels($\mathbf{\Phi}$, $f$)              $\triangleright$ Get all voxels that fascicle $f$ passes through
4:      vc $\leftarrow$ GetFirstVoxel(seen_v, $\mathbf{v}$)      $\triangleright$ Select a voxel that has not been seen for fascicle $f$
5:      seen_v $\leftarrow$ seen_v $\cup \{$vc$\}$
6:      an $\leftarrow$ PlotFirstVoxelNodes(vc, seen_v)   $\triangleright$ Plot the nodes of the current voxel and return the next voxel
7:      **repeat**
8:          an $\leftarrow$ PlotNodes(an, seen_v)
9:      **until** an is null
10: **procedure** PLOTFIRSTVOXELNODES(vc, seen_v)
11:      anset $\leftarrow$ AllNeighbouringNodes(vc, seen_v)       $\triangleright$ Get all active nodes in the neighbouring active to vc
12:      $\mathbf{acset} \leftarrow$ non-zero($\mathbf{\Phi}$(:,f, vc))              $\triangleright$ Get all active nodes in the current voxel
13:      ac, an $\leftarrow \mathrm{argmin}_{an' \in \mathbf{anset}, ac' \in \mathbf{acset}}$(AngularDistance(an', ac'))     $\triangleright$ Finds the closest nodes between $\mathbf{acset}$ and $\mathbf{anset}$
14:      astack $\leftarrow$ empty stack
15:      **while** $\mathbf{acset} \neq \emptyset$ **do**
16:          push ac onto astack
17:          $\mathbf{acset} \leftarrow \mathbf{acset} -$ ac                       $\triangleright$ Remove ac from $\mathbf{acset}$
18:          ac $\leftarrow \mathrm{argmin}_{ac' \in \mathbf{acset}}$(AngularDistance(ac, ac'))     $\triangleright$ Find the next closest node to the previous node in the current voxel
19:      **while** astack not empty **do**
20:          ac $\leftarrow$ pop astack
21:          Plot(ac)                             $\triangleright$ Pop the nodes from astack and plot them
         **Output:** an, The closest node to the last plotted node
22: **procedure** PLOTNODES(ac, seen_v)
23:      vc $\leftarrow$ voxel containing ac
24:      $\mathbf{acset} \leftarrow$ non-zero($\mathbf{\Phi}$(:,f, vc))              $\triangleright$ Get all active nodes in the current voxel
25:      **while** $\mathbf{acset} \neq \emptyset$ **do**
26:          $\mathbf{acset} \leftarrow \mathbf{acset} -$ ac                           $\triangleright$ Remove ac from $\mathbf{acset}$
27:          Plot(ac)                               $\triangleright$ Plot the current node
28:          ac $\leftarrow \mathrm{argmin}_{ac' \in \mathbf{acset}}$(AngularDistance(ac, ac'))     $\triangleright$ Find the next closest node to the previous node in the current voxel
29:      anset $\leftarrow$ AllNeighbouringNodes(vc, seen_v)       $\triangleright$ Get all active nodes in the neighbouring active to vc
30:      an $\leftarrow \mathrm{argmin}_{an' \in \mathbf{anset}}$(AngularDistance(an', ac))   $\triangleright$ Finds the closest node in $\mathbf{anset}$ to the last plotted node
         **Output:** an                               $\triangleright$ The closest node to the last plotted node

**Algorithm 5** Minimum angular distance metric

---

**Input:** Two three dimensional sparse tensors $\mathbf{\Phi}_{\text{exp}}$ and $\mathbf{\Phi}_{\text{pred}}$, the expert and predicted brain structures.

1:  total_dist $\leftarrow 0$
2:  **for** $v_p = 1, \ldots, N_v$ **do**
3:     dist_per_voxel $\leftarrow 0$
4:     ap_set $\leftarrow \mathcal{P}(\{a | a \in \mathbf{\Phi}_{\text{pred}}(:, v_p, :)\})$ ▷ The power set of all orientations active in the current voxel
5:     **for all** $a_{\text{exp}} \in \mathbf{\Phi}_{\text{exp}}(:, v_p, :)$ **do**
6:        $\text{vec}_{\text{exp}} \leftarrow a_{\text{exp}} \times \sum_{f_i \in \mathbf{\Phi}_{\text{exp}}(a_{\text{exp}}, v_p, :)} \mathbf{\Phi}_{\text{exp}}(a_{\text{exp}}, v_p, f_i)$
7:        min_dist $\leftarrow \infty$
8:        **for all** $s \in$ ap_set **do**
9:           $\text{vec}_{\text{pred}} \leftarrow \sum_{a_i \in s} a_i \times \sum_{f_i \in \mathbf{\Phi}_{\text{pred}}(a_i, v_p, :)} \mathbf{\Phi}_{\text{pred}}(a_i, v_p, f_i)$
10:         distance $\leftarrow$ Angular_Distance($\text{vec}_{\text{exp}}, \text{vec}_{\text{pred}}$)
11:         **if** distance $<$ min_dist **then**
12:            min_dist $\leftarrow$ distance
13:        dist_per_voxel $\leftarrow$ dist_per_voxel + min_dist
14:     total_dist $\leftarrow$ total_dist + dist_per_voxel
     **Output:** total_dist           ▷ The total angular distance

(a)

(b)

Figure 8: **(a):** Distribution of reconstruction error over voxels for each initialization strategy for orientations on the **ARC-SLF** dataset. **(b):** The improvement of reconstruction error during the steps of gradient decent shows that the objective is not able to improve the OMP selected orientation sets while it is improving the GreedyOrientation choices constantly.

(a) Ground truth            (b) OMP            (c) Greedy

Figure 9: The quality of orientation sets selected in screening stage comparing to the ground truth orientations on the **ARC-SLF** dataset. **(a)**: Initializing $\underline{\mathbf{\Phi}}$ with expert $\underline{\mathbf{\Phi}}$, **(b)**: Initializing $\underline{\mathbf{\Phi}}$ with OMP, **(c)**: Initializing $\underline{\mathbf{\Phi}}$ with GreedyOrientation.

(a) Worst-GreedyOrientation-initialization

(b) best-GreedyOrientation-initialization

(c) Worst-OMP-initialization

(d) best-OMP-initialization

Figure 10: Top five best and worst fascicles for OMP and GreedyOrientation after optimization according to reconstruction error. Solid lines show the predicted $\underline{\Phi}$ while dashed lines are the ground truth. The predicted $\underline{\Phi}$ are of a different scale than the ground truth, making direct comparison difficult; however, the structure and shape of the fascicles in **b** clearly align closely with their ground truth counter parts.