[Reviews · NeurIPS 2019]

Reviewer 1



- The paper builds on ENCODE, formulating the connectome decomposition problem as a tensor product of \Phi and \D, where \D is a dictionary of dMRI signal predictors. Though this framework is taken from a previous publication, it would be nice to include at least a brief introduction and summary of this D -- how was it constructed, what do its elements represent, and what's the intuition behind it? - What is this dimension N_{\theta}? It first appears in Section 2 with no definition. - How was the regularizing parameter \lambda chosen in the empirical example? How sensitive is the algorithm to choice of this parameter?

Reviewer 2



# Overview This paper proposes a new technique to reconstruct entire white matter fasciculus directly from diffusion magnetic resonance imaging (dMRI) data. Moreover, that technique doesn't rely on an iterative process to generate the fascicles as classic tractography algorithms do. Instead, it learns a sparse 3D tensor that encodes the location (i.e. voxel) and orientation of all (relevant) white matter fascicles in a brain subject. Fitting such tensor can be done with gradient-based optimization. However, because it has so many parameters (i.e. ~10 billion for 823 fascicles, 1057 discretized orientations, and 11,823 voxels) and it is a highly sparse problem, the authors propose to use a screening algorithm to select a plausible set of orientations for each voxel based solely on diffusion information. To do so, they propose a modification of the Orthogonal Matching Pursuit (OMP) algorithm where the orthogonality constraint is relaxed in favor of having high multiple correlations and ensuring each orientation itself is useful at explaining the diffusion information. In addition, the author also proposes a group sparse regularizer to ensure biologically plausible fascicles, i.e. locally, fascicles should be smooth and continuous. The authors validate their approach to two major white matter structures in the brain that were extracted by a connectome expert. They show the proposed technique can learn those two structures efficiently and results in low reconstruction error with smooth fascicles. The authors also show their proposed screening algorithm is more suitable than the original OMP algorithm in this context. Overall, I found the paper is well written, the proposed technique interesting and the motivation is clear. To the best of my knowledge what's proposed in this paper is novel. Moreover, the authors do mention this is preliminary work and focused on providing a sound formulation and providing an initial empirical investigation into the efficacy of the approximations. # Major concerns Even though the authors claim it is efficient, I wonder how well the proposed method scale with the number of tracts and voxel resolution? Traditional whole-brain tractography can easily have many hundreds of thousands of tracts. That said, one could easily argue those techniques oversample the number of tracts. I'm curious to know if the authors have tried to learn a smaller Phi (i.e. reducing the size of the 3rd dimension) and see at what point there aren't enough tracts to properly explain the diffusion signal. Visualizing the resulting Phi is essential for any practical use of this technique. As mentioned in the Appendix, it requires a whole other optimization process in order to place and connect the segments of the tract together. I'm concerned about how reliable that process is, and what impact the greedy algorithm has on the result. # Minor concerns From the main paper, it is not clear how you limit the tractography to only a particular fasciculus. From Algorithm 3 in the Appendix, it seems you have a set of masks. From the text, it is not clear how D is obtained. In Figure 6, it is not clear how corresponding ground truth tracts were determined. If I understood correctly, the solution obtained for Phi could have permutation in its 3rd dimension compared to the ground truth. So, to establish a correspondence between a predicted tract and the ground truth, did you rely on the lowest reconstruction error? Since source code availability is not mentioned in the paper (but included in supplementary), I'd invite the authors to release their code. # Typos line 238 and 239: l1 -> $\ell_1$

Reviewer 3



Overall this work has multiple original contributions, the main one being the formulation of tractography estimation as a fully unsupervised learning problem. While the base of the formulation is not particularly novel (minimization of a linear least-squares objective via subgradient descent), the authors introduce a convex regularizer that encourages biologically plausible solutions through continuity in space and orientation of learned fascicles, which is a novel addition to help yield better solutions specifically in this domain. Furthermore, the use of a custom objective for screening orientations for each voxel by greedy selection is an original contribution that seems crucial for achieving plausible results as demonstrated by the experiments. The work is clearly written in general, with each component explained well. One comment is that there seems to be no explanation for the choice of the regularization parameter in Section 5.2; it would be worthwhile to justify this choice, or explain a proposed method for choosing this parameter in general. The overall technical quality is good, with convincing empirical evaluation and theoretical claims that are elaborated in the Supplement and appear to be sound. However, one primary shortcoming of the work is that there is no indication of how the specific choice of convex regularizer affects the results, and the empirical evaluation seems to indicate that the combined effect of the greedy selection method and the regularizer is necessary for the learned fascicles to be continuous (i.e., convex regularizer + OMP still leads to subpar results, indicating that the convex regularizer alone may be insufficient). Another main component that is lacking was brought up by the authors themselves in the Discussion: "understanding strengths and weaknesses compared to current tractography approaches". An empirical comparison to alternative tractography estimation methods would validate the unsupervised learning approach overall, and not just the use of a custom greedy selection method for screening. Since the missing components described above seem crucial, I lean towards rejecting the submission, though the inclusion of either component would lead me to increase my score. Other minor comments: - Line 228: seems like this should be "Figure 5a" - Line 231: seems like this should be "Figure 5b" - Lines 251-252: Figure 6 references have typos Update based on author feedback: Having read the authors' response, I feel that my concern about the effect of the custom regularizer versus the greedy screening method was appropriately addressed, and I no longer consider this an issue. I still feel that a lack of any kind of comparison to other tractography methods is a main shortcoming of the paper, despite the authors' claim that this is outside the scope of this submission. However, the authors' response has led me to lean more towards accepting the paper overall, and I have updated my score to reflect this.

[Author Response · NeurIPS 2019]

**Regularization**

Finding an appropriate value for hyper-parameters is always a challenge in machine learning problems. To select an appropriate regularization parameter, we want to (a) satisfy biological plausibility conditions of fascicles as well as (b) reconstruct the diffusion signal Y. We started with a larger regularization parameter and decreased it until we found the reconstruction error did not change much, to find a suitable balance. We found that picking this parameter was not too difficult, though tuning it more would likely result in better performance.

**Prior Tractography Results**

We do not yet compare to prior tractography algorithms. Rather, we first ask: how effective is our new formulation and algorithms, given ground truth? We use LiFE, which is a tractography evaluation method, with one tractography solution to give us ground truth. The availability of a ground truth Phi let's us focus our investigation into the soundness of the formulation and the algorithms. The computational complexity of solving the formulation is non-trivial, and understanding the soundness of the proposed algorithms is critical before deploying at a larger scale. Introducing this new formulation and developing and understanding sound approximation algorithms for this formulation is the right size for a NeurIPS paper; also demonstrating performance for full tractography solutions, across brains and against many other tractography methods, would be large journal paper let alone NeurIPS paper. The next step, though, is absolutely to do this comparison.

**Problem Formulation**

Reviewers 1 and 2 both asked how the dictionary $D$ was obtained and Reviewer 1 asked about $N_\theta$. dMRI measurements are collected with and without a diffusion sensitization magnetic gradient and along several (i.e. $N_\theta$ number of) **gradient directions** ($\theta \in R^3$). The data was collected for $N_\theta = 96$ different angles of gradient direction. Then matrix $D \in R^{N_\theta \times N_a}$ is a dictionary of diffusion predictions with the same type as diffusion signal, whose columns (i.e. $N_a$ atoms) correspond to precomputed fascicle orientations, and intuitively shows that what diffusion signal we expect to receive from any axon in the direction of any possible fascicles orientations ($a$) in space by sensitizing magnetic gradient in each direction of ($\theta$). More specifically, the entries of the dictionary were computed as follows: $D(\theta, a) = e^{-b\theta^T Q_a \theta} - \frac{1}{N_\theta} \sum_\theta e^{-b\theta^T Q_a \theta}$, in which $Q_a$ is an approximation of diffusion tensor per fascicle-voxel. $\theta^T Q_a \theta$ gives us the diffusion at direction $\theta$ generated by fascicle $f$. We will include this information in section 2 of the paper for the final draft.

**Reviewer 1**

With the additional space in the camera-ready, we can include more background and discussion on ENCODE.

**Reviewer 2**

Our solution exploits the inherent sparseness in the optimization. Each voxel has a very small subset of potential fascicles and orientations. The algorithm scales well with an increase in the size of the tensor. The greedy algorithm to select orientations does have to iterate over all orientations once, for each voxel, before the optimization is run. However, this does not need to be run for each step of the optimization and is not expensive to do once upfront. The optimization then scales effectively linearly with the number of voxels considered, because a relatively small maximum number of orientations is considered for each. Additionally, the optimization is very easily parallelizable, allowing our approach to scale with the amount of available computational resources.

We did not include results across different subjects due to space constraints. We can include a few more subjects in the appendix of the final draft.

The greedy algorithm used for visualization has been carefully designed to reflect biological attributes of fascicles such as smoothness and continuity. It finds the best possible permutation of the segments for a fascicle in each voxel-based (a) on the two-by-two angular distance of each segment's orientations and (b) neighboring voxels. We believe the algorithm is reliable, both because of the principle in its design and because we extensively visualized several ground-truth and learned solutions to find any issues. You are right that we cannot guarantee that it provides a perfect visualization, and there is more to be understood about how to faithfully visualize our solutions.

We will include details about how fascicles are screened in the main text.

**Reviewer 3**

The main concern here is that the regularizer + OMP does poorly, suggesting the regularizer on its own is inadequate. However, this is a slight misunderstanding. OMP and GreedyDirection both first screen the set of considered orientations. This introduces an approximation, to make the optimization much faster (so that it is a feasible optimization). The best for the regularizer would be to do no screening. The ordering would be "Regularizer Alone" is better than "Regularizer + GreedyDirection" better than "Regularizer + OMP". The fact that "Regularizer + GreedyDirection" does well suggests the regularizer is actually reasonable and that we might even be able to better if we can improve the orientation screening. We only include OMP to demonstrate that we needed a better screening method than existing methods (i.e., OMP).

[Meta-Review · NeurIPS 2019]

The reviewers have reached the decision to recommend acceptance based on the authors' clarifications in their feedback. I highly recommend the authors to take these thoughts into consideration in editing the paper for the camera-ready.